# A Multi-Flow Production Line for Sorting of Eggs Using Image Processing

**DOI:** 10.3390/s23010117

**Published:** 2022-12-23

**Authors:** Fatih Akkoyun, Adem Ozcelik, Ibrahim Arpaci, Ali Erçetin, Sinan Gucluer

**Affiliations:** 1Department of Mechanical and Metal Technologies, Trabzon Vocational School, Karadeniz Technical University, Trabzon 61300, Turkey; 2Department of Mechanical Engineering, Faculty of Engineering, Aydın Adnan Menderes University, Aydin 09010, Turkey; 3Department of Software Engineering, Faculty of Engineering and Natural Sciences, Bandırma Onyedi Eylul University, Bandırma 10200, Turkey; 4Department of Naval Architecture and Marine Engineering, Faculty of Maritime, Bandırma Onyedi Eylul University, Bandırma 10200, Turkey

**Keywords:** industrial egg grading, egg weight estimation, image processing, computer vision, multi-flow production line

## Abstract

In egg production facilities, the classification of eggs is carried out either manually or by using sophisticated systems such as load cells. However, there is a need for the classification of eggs to be carried out with faster and cheaper methods. In the agri-food industry, the use of image processing technology is continuously increasing due to the data processing speed and cost-effective solutions. In this study, an image processing approach was used to classify chicken eggs on an industrial roller conveyor line in real-time. A color camera was used to acquire images in an illumination cabinet on a motorized roller conveyor while eggs are moving on the movement halls. The system successfully operated for the grading of eggs in the industrial multi-flow production line in real-time. There were significant correlations among measured weights of the eggs after image processing. The coefficient of linear correlation (R^2^) between measured and actual weights was 0.95.

## 1. Introduction

Large-scale production provides the mass production of goods with the help of production line processes and advanced technologies as a basis [1]. Production lines allow manufacturers to draw advantage from economies of scale, producing more products with lower costs [2,3,4]. In general, manufacturers are investing in automated production lines to take this advantage. Recently in many countries, small-scale factories have been transforming their production lines into automated solutions. Mostly, automated industrial grading processes include a Computer Vision System (CVS) because it provides swift calculations and accurate results in a short time [5] for detecting and discriminating agricultural products by their size and shape [6], dirt rate on production lines [7,8,9].

In many countries, the size of eggs is the primary parameter for classifying eggs [10]. In the market, size is an important factor determining consumer preferences [11,12,13]. The size of eggs is also a determining factor on the final taste and flavor, especially in bakery products. Using different sizes of eggs affects texture, flavor balance, consistency, and mostly gives an unsatisfactory impression for the consumers [14].

Specific requirements for produced eggs are stated in European egg size standards [15,16]. The main parameter of an egg is the weight for grading, which should be within a specified range (Table 1) for both the producer and the consumer, constituting the quality criteria for eggs.

In agri-food industry, sorting is playing an important role [17,18,19] in increasing the value of the products in the market and marketing capability [20,21,22]. The sorting process is providing uniformity in size and shape, decreasing the costs related to the packaging and transportation, and offering the most ideal packaging configuration [23,24,25]. Most of the conducted studies agree that the accuracy of a vision based egg grading process depends on imaging resolution, sorting methods, and the color of the eggshell [26,27,28]. Although most of the studies [29,30] use clean eggs for egg size determination process, in this study both clean and dirty eggs were used as work material. In the experiments, four grades of eggs from small to large were examined respect to the European egg grading standard [15]. In the egg-related food industry, there is a need for rapid and safe digital classification of eggs using image-processing technology.

Image processing applications use images or video streams. An encoded image such as JPEG, GIF, and PNG has too many digital numbers stored in it. A bitmap with three channels consists of 24-bit data for every pixel of an image with 8 bits of RGB color for each channel [31,32]. An obtained image from the vision sensors is the reflectance of a light source from the surface of an object, which is recorded pixel by pixel. Moreover, the computerized vision sensors used in image processing applications focus on the reflectance of the target measurement area rather than the target object [33]. For this reason, surface illumination of the processing area on the production line is a very important parameter for increasing the accuracy performance of the image processing application. In CVS applications, besides many advanced software solutions for classifying products by their size, color, dirt, and shape [34], other important parameters are also considered such as light sources for illumination, spectral reflectance of the ground [35], imaging resolution of the vision sensor, image and data processing speed of the Central Processing Unit (CPU) and Graphical Processing Unit (GPU) [36,37]. Most of all, the illumination of the measurement area is one of the dominating parameters for the CVS applications [38].

Today, in addition to advanced technology, an illumination cabinet with proper light source and layout is a crucial factor for MVS based multi-flow production lines. Using insufficient lighting is decreasing the prediction accuracy of the CV software since it causes shadow and saturation problems. Further, over-illumination is causing saturation due to the overlapped light sources. Mostly, it is seen at the central region of the MVS production line. To increase the accuracy of the CV software, multiple light sources with adjustable intensity are a very important parameter for providing smoother lights on the object without reflection of background from the measurement area especially for the MVS based multi-flow production lines. Most of the study in the literature focuses on a single object with a single light source for measuring and predicting parameters of a target object. In general, most of the proposed methods offer complex algorithms that require high-cost computer hardware solutions for reducing the noise caused by the light sources. Typically, these solutions are proven using a single object with single light source. There is no fully achieved solution on light noise reductions for both single and multiple objects. It is important to investigate the effect of multi-light sources on CV software, especially for real-life problems in MVS-based industrial production lines.

## 2. Related Studies

Chicken is the most widely grown poultry species worldwide [39]. The poultry industry is often divided into meat and egg production segments [40]. With approximately 529 billion eggs, China is the leading egg producing country, followed by the US, India, Mexico, and Brazil [39]. Manual and automatic systems are used to classify eggs by size. A new technique is also used to make the classification process of eggs accurate, fast, economical, and efficient. Computer vision technology has proven to be sufficient to achieve this goal.

In the poultry production industry, computer vision techniques have been used for egg grading and sorting [41,42]. For instance, Valdez et al. [43] developed an automated system for sorting fertilized duck eggs in terms of gender. Using a microcontroller interface, duck eggs were segregated according to gender based on their shapes. The results indicated that prediction accuracy of the proposed egg sorting system was 91%. Alikhanov et al. [25] sorted chicken eggs into four classes based on their weights by employing regression analysis. There is a relationship between weights and geometric parameters such as area, minor and major axis, shape factor, and index. The classification errors were 2.5% for testing and 12.5% for training sets. In another study, Alikhanov et al. [44] developed an automatic system for sorting eggs by shape assessment and indirect weight using computer vision. Geometric parameters such as major and minor axis, perimeter and area, shape factor, and index were attributes of the algorithms. The results indicated that sorting accuracy was 94.6% and 90.3% for testing and training, respectively.

Priyadumkol et al. [45] developed a computer vision system for crack detection in eggs based on image processing. The results, which were validated by using 750 egg images, indicated that prediction accuracy was 94%. In another study, Wu et al. [46] aimed to detect cracked and intact eggs using transmission imaging along with SVM classifier. The results indicated that the accuracy in testing and training sets were 93% and 94%, respectively.

In the literature, there are limited studies on determining the volume and weight of chicken eggs. Many studies have been performed by image processing of the egg in a still position under a single light source [16,47,48,49]. However, in batch processing single flow lines and still image-based approaches are not of concern. In industrial applications, the importance of multi-flow egg grading is essential to obtain speed. Typically, the conventional multi-flow line egg grading applications use load-cell dependent measurement methods. It is a low-cost solution but there are limitations such as wrong measurement due to the cracked eggs on the cells. In addition, eggs should be stay stationary when measuring the weights due to the mechanical force measurement.

This study offers an industrial multi-flow grading MVS using a standard RS camera by processing real-time images. An MVS is expressed by providing a competitive, fast, and accurate setup using a standard low-cost RS camera. The demonstration of a real-life and real-time application is achieved using a low-cost real-time methodology on multi-flow industrial roller conveyor line. The study provides an economic and reliable option due to the offering of a non-contact measurement method for multi-flow egg grading lines for real-life applications.

## 3. Materials and Methods

Production lines with CVS mainly consist of three major units to implement sensing, processing, and acting duties. To implement these duties, the CVS applications require electronic hardware, mechanical actuators, and image processing software. To meet these requirements, a Red Green Blue (RGB) colored camera is a common solution at the sensing stage. A personal computer (PC), industrial PC (IPC), a dedicated microcontroller unit (MCU), or a Field Programmable Gate Array (FPGA) is utilizable as the main controller for processing images. In the acting stage, an MCU-based electronic board is mostly chosen option to provide communication between the controller and acting mechanism [50]. For an industrial production line, which consists of CVS, the necessary components are a conveyor band, a light source, programming, an image-processing library, MCU boards, a camera, and a PC for processing the image data.

Typically, the CVS uses image processing techniques for detecting and classifying the products. To achieve these goals, there are many languages, software, and libraries. Open Source Computer Vision Library (OpenCV) is a good option for image processing applications [51]. It is suitable for advanced solutions, supports real-time vision applications, and has interfaces for programming languages such as C++ which is a general-purpose programming language [52,53]. Moreover, it includes many functions for processing images and implements swift calculation with the help of C++ for processes such as contour finding, image thresholding, and resizing.

### 3.1. Experimental Setup

In this study, the layout of the egg movement halls was demonstrated in Figure 1. The columns vertically divided by four parallel lines based on the multi-source illumination. A row is selected as Region of Interest (ROI) for processing eggs concerning the flow direction. An isolated illumination cabinet, which was manufactured by referencing the dimensions of a roller conveyor mechanism with egg movement halls, was mounted on the production line. An imaging device with adjustable LEDs (4 W), total of 120 pieces with 15 rows and 8 columns, was integrated into this cabinet. In the experiments, the LEDs were placed according to bright field illumination technique. The light source was placed to the top side of the cabinet and each flowline was illuminated using a series of LEDs with equal spacing. In the experiments, an adjustable power source was used to supply the LEDs which was adjusted to 12 V and the LEDs were drawing 0.33 A current, during experimental measurements

An industrial computer (IPC) with Intel i5 Central Processing Unit (CPU) @ 2.50GHz was used to obtain images from the imaging device over the Universal Serial Bus (USB). The PC was used to acquire imaging data from the device at 30 fps with 1280 × 720 resolution.

### 3.2. The Illumination Cabinet

This research was conducted in the industrial environment with the help of the isolated illumination cabinet. A USB-supported image acquisition device was integrated into this cabinet. For image acquisition, a color camera model was used, which had a CMOS optical sensor and FoV as 78°. The illumination cabinet was divided four parts concerning egg flow lines and four LED arrays for four columns were used to illuminate these flow lines. Each LED array was connected to a potentiometer for adjusting the intensity of the light source. The light sources and the imaging device are shown in Figure 2.

### 3.3. Image Processing Procedure

Object detection and classification steps were conducted using the OpenCV image processing library and C++ language using a personal computer (PC). The imaging module was connected via USB to the PC for accomplishing imaging acquiring duty. Images of the target objects were obtained from the imaging device using the OpenCV library and C++ programming language. Image processing procedure for detecting and weight predicting processes are indicated in Figure 3.

Each measurement was conducted by reference to the given processing method. The process initially converts the actual image from Red Green Blue (RGB) channels to Hue Saturation Value (HSV). The normalized box filter and the Gaussian filter are applied to the image streams for blur filtering operation to smoot the image streams. HSV image and a masking process is applied to obtain the actual image but background by removing background stage which is conducted with the help of range operation. The masked image with a thresholding operation is used to detect object contours. In the next stage, minimum enclosing circle method was applied (Figure 4). The minimum enclosing circle is a circle in which all the points lie either inside the circle or on its boundaries [54]. This process is used to increase the accuracy of size determining process especially for dirty eggs. The dirtiness on eggs affects the measurement when it reaches the borders of the eggs. This error is minimized if the ellipse is circulated respect to the outer circle.

In the last stage, the data including black and white pixels are used to determine zero and non-zero-pixel counts. The ratio of the determined contour area and the zero-pixel rate is the result output of the CV software, and the output is used to grade eggs by their weights.

The detected eggs evaluated by reprocessing ROIs with filtering, thresholding, and pixel counting operations. The detected area size was used for predictions and the predictions were calculated concerning BW pixel ratio in the detected ROIs. A sample result from the process is shown in Figure 5 for ROI of the actual image (a), the threshold of the grayscale image (b), mask (c), and the detected egg (d).

The supervised learning method is applied using the contours parameters such as area, perimeters, and position that obtained from the detected eggs with image processing procedure. The processed image results are evaluated in real-time. The computer vision output parameters were used in supervised learning process as labeled data. In the training stage, 67% of the data were used for training and rest were used for testing process.

### 3.4. The Measurement Cabinet

Typically, commercial cameras are used in the passive mode that require an external power source to illuminate targeted objects. In factory conditions, using daylight illumination is decreasing the performance of the CVS because of the input source is continuously varying from day by hours. A pre-conditioned environment, which has a constant light source for illumination provides better results for object classifying applications in egg production lines. Moreover, for active sensor systems, which use internal power sources to sense an environmental change, the proper illumination becomes a dominating factor in object classifying applications. In the visible spectrum, poor illumination causes false positives and abnormal illumination incurs saturation problems [8]. Adjustable light sources solve a part of the noises, but using multiple light sources provides smoother illumination on the objects, which aid to reach high accuracy in object classifying applications. In image processing applications, surface illumination of the processing area for a production line directly affects the performance. For preventing from saturated and under illuminated regions at the surface of the processing area, using multiple light sources such as LED series assure more smoothly illuminated regions instead of using a high-powered single beam light source. In Figure 6, independent multiple light source (Div 1 to Div 4) reflectance on the measurement ground is demonstrated in images.

In image processing applications, artificial lighting is needed for dim or dark conditions [34]. For this reason, illumination plays an important role in the accuracy of the measurement system. Under or over-illumination both produce noise on the spatial resolution of the image data. There are shadows related to the under illuminated regions and saturation is a big problem because of the over illuminated regions. The typical solution for increasing the accuracy of the image processing applications is minimizing the noises at the input with proper light sources and providing smoothly reflected regions rather than sharply illuminated regions. In addition to the specific properties of the image processing systems, the light sources used for the illumination of the measurement area play significant roles in image processing applications, especially for multi-flow production lines.

### 3.5. Determining the Field of View (FoV)

For this study, *FoV* is an important parameter for determining the line bounding of the multi-line flow production line. It should be investigated for dividing the camera vision to proper bounds. For a rectangular image located in front of the optical axis, three angular parameters of the *FoV* are of concern: horizontal, vertical, and diagonal [55]. Typically, imaging device manufacturers use diagonal *FoV* (*DFoV*) to define the *FoV* refers to the visual angle of a camera lens. Horizontal *FoV* (*HFoV*) describes horizontal dimensions and vertical *FoV* (*VFoV*) describes vertical dimensions of the visual field. For calculating the field of view (*FoV*) of a camera, the relationship between the focal length of the device and sensor size was considered (Figure 7).

The *FoV* (α) was calculated using Equation (1), which refers to the image plane dimension (*w*) and the focal length (ƒ), which is the distance between lens and image plane.
(1)$$\alpha =2\;\tan^{-1}\frac{w}{2f}\;\mathrm{or}\;f=\frac{w}{2}\cot\frac{\alpha }{2}$$

Based on these equations, using the image width *ω*, the field of view *α* and the focal length *f* can be calculated. The angles of the camera sensor were 70.42° and 43.30° for *HFoV* and *VFoV*, respectively, as indicated in Figure 8. The angle was 78° with a 16:9 aspect ratio for *DFoV*.
(2)$$DFoV=2\arctan\left(\frac{d}{f}\right)$$
(3)$$HFoV=2\arctan\left(\frac{w}{f}\right)$$
(4)$$VFoV=2\arctan\left(\frac{h}{f}\right)$$
where *d* is the diagonal length, *w* is the width, and *h* is the height of the *FoV*. The *FoV* can be determined by multiplying *HFoV* with *VFoV* and the aspect ratio of the frame is very important as well as the focal length, which is the distance between the image plane and the camera lens. The *FoV* is used to determine the distance between the camera and measurement area, which is the height of the illumination cabinet. The width and length of the cabinet were determined concerning *DFoV*. The determined *FoV* for the imaging device was divided into four separate columns and each column was named up to four from left to right by referencing the top view regarding the denoted calculations.

### 3.6. Evaluating the Accuracy of the Measurement System in Real-Time

In the preliminary test, for evaluating the accuracy of the measurement system, the areas of the four identical samples were predicted for single and multi-source illumination. The actual areas of the samples were measured using a precision caliper and compared with the predicted area. The original image of the identical objects (a) and the processed image (b) to detect objects was shown in Figure 9.

### 3.7. Experimental Procedure in the Multi-Line Flow Egg Production Line

The experiments were conducted in the industrial motorized roller conveyor with four lines of egg movement halls. In the experiments eggs graded (Figure 10) by their weights. The weight of the eggs was determined using a sensitive scale with 0.01 precision. Each egg was transferred by movement halls and detected via CVS using an identical image processing method. The results of this experiment were used for evaluating the effect of illumination on the image processing procedure for egg weight determining and grading in industrial multi-flow line.

The egg weight measuring step was only conducted when an egg entered in the ROI area. Each datum for detected eggs was predicted when the mass center of the egg was passing the input line. Real-time egg detection and dimension predicting processes were accomplished for single and multi-flow lines on industrial egg movement halls with custom-designed cabinet.

## 4. Results and Discussion

Classification is an inevitable process in the agri-food industry to increase the marketing capability and marginal product value by saving time and reducing the costs [56,57]. Today, with the help of advanced technology, researchers are focusing on image processing techniques for grading products mostly by their size and shape [58,59]. Accordingly, in a study conducted by [41], it was found that undesirable stains affected the pixel numbering and caused a high error in size determination. Dirt, breakage, and cracks on the egg cause this error and they used the infill function for avoiding from this error but there are limits when the stains are located at the boundary of an egg. Accordingly, under-illumination causes this problem because of similarities of the shadows and stains for pixel numbering.

In this study, real-time image processing accuracy was evaluated for multi-line flow and averaged processing speed measured. The resolution was 1280 × 720 pixels of captured images from the camera stream. The average processing speed was observed (Table 2) around 0.092 s per frame notwithstanding object count.

Experimental results were evaluated in three stages by conducting real-time measurement on an industrial motorized roller conveyor with egg movement halls. At the first stage of this study, to maximize the accuracy of the predictions and to prevent the noise of the images, the image processing method and illumination cabinet were tested on a single-line flow with graded eggs. For comparing the effect of the single and multi-line flow, the areas of identical objects were predicted using the same image processing method and cabinet. In the final multi-line flow experiment, multiple eggs were tested under multi-source illumination for four egg movement halls on the roller conveyor (Figure 11).

Single and multi-line flow experiments were conducted with motorized egg movement halls in real-time. Single-line flow test was conducted for four graded sample eggs one by one, and an example measurement image is shown in Figure 12. While a single egg (a) was entering the imaging area, the mass center of the egg was detected (b) by the MVS and weight calculated when the egg was in the ROI. A multi-line flow experiment with multiple eggs is shown in Figure 12 as an obtained image (c) and detected eggs in the processed image (d).

### 4.1. The Accuracy of the Measurement System

The accuracy of the measurement system was evaluated for single and multi-light source illumination. Identical objects were measured for both single and multi-light sources. Four identical objects were used with the opaque black ground for evaluating the accuracy of the image processing procedure. The identical objects (Figure 9) were placed at the target location in the *FoV* of the camera concerning the four flow lines. The measurement result for each flow line of the identical objects was shown in Figure 13, both for single and multi-line-based illumination. The error percentage of the predicted area for a single light source was around 2%. In addition, with multi-source illumination, flow lines give better results and error variations decrease up to 0.6% for static measurements.

### 4.2. Measurements of Graded Eggs for Single Line Flow in Real-Time

In this step, the weights of four different graded eggs were predicted for one flow line for constant flow speed at 0.2 m/s. The single-line flow test for predicting the weight of eggs was conducted for four graded eggs with four-time repetition. The conducted measurement showed (Figure 14) that predictions were accurate for different sizes and the relationship between size and weight gave a high correlation. The predicted and actual weight coefficient of linear correlation (R^2^) was 0.99. Error bars in the result are the standard deviations of the minimum four different conducted measurements.

### 4.3. Measurements of One Egg for Multi-Line Flow in Real-Time

In the next step, the weight of one egg was predicted for four flow lines. The sample egg was used to collect data from each flowline on the movement halls. The egg was rolled via the motorized roller conveyor for each line at 0.2 m/s constant speed and its weight was predicted with CVS regarding the same image processing procedure. Obtained results from the multi-line flow with multiple light source illumination are shown in Figure 15. Error bars indicated the percentage of the error concerning the actual weight value. The test results showed the variances were under 1.6% using multi-source illumination in real-time.

### 4.4. Measurements of Four Graded Eggs for Multi-Line Flow in Real-Time

In the last stage of the study, the test eggs were rolled from four flow lines and each egg measurement was repeated four times. Results were obtained with the percentage error of the area predictions for four flow lines of the multi-flow egg production line and multiple eggs. Figure 16 shows the results from four flow lines and the correlation between predicted and actual weight of the eggs which was close to 0.95. The measurements showed all classes have standard deviations under 3%. Moreover, the measurement accuracy results to grade S, M, L, and XL (Table 1) class eggs were above 95%.

Most of the studies in the literature have been carried out on still images for detecting and determining the size and weight of eggs [16,47,48,49]. In these studies, measurements were made by fixing the egg on the floor to be imaged and the success rates were obtained as 87.6% by Thipakorn et al. [49], and 95.0% by Cen et al. [16]. However, in industrial egg production facilities for bulk production, it is required to measure egg parameters in real-time. In the present study, the measurements were carried out while the eggs were in motion on a roller conveyor mechanism and results were obtained for multi-line flow are more successful than those in the literature.

## 5. Conclusions

In practical/industrial applications, it is inevitable to process multiple eggs with multi-flow production lines. Most of the studies in the literature were carried out on still images for detecting and determining the size and weight of eggs. In the present study, an image processing technique with a multi-source illuminating cabinet was applied accurately to predict egg weights on a multi-flow egg production line in real-time.

The system successfully operated for the grading of eggs. Although measurements were carried out while the eggs were in motion, the error was below 5% to grade S, M, L, and XL class eggs for multi-line flow with multiple eggs in real-time. In addition, it was demonstrated that the accuracy of the image processing method could be increased with the help of multiple light sources.

The predictions of the egg weights concerning egg area gave a high correlation with the actual egg weights. The accuracy of measurements respect to the area and actual weights was found to be 99%. The identical objects measurements, under multi-light source was found around 0.6%, but for the single light source measurements, the error was found over 2%.

The proposed computer vision system can be used by manufacturers for sorting eggs by weight and can be an insight for future studies dealing with sorting of food commodities. It is foreseen that the measurements will be carried out quickly and successfully without the need for the use of scales for weight determination in egg production facilities. Therefore, there is an increase in packaging speed with the help of advanced machine vision systems.

## Figures and Tables

**Figure 1 sensors-23-00117-f001:**
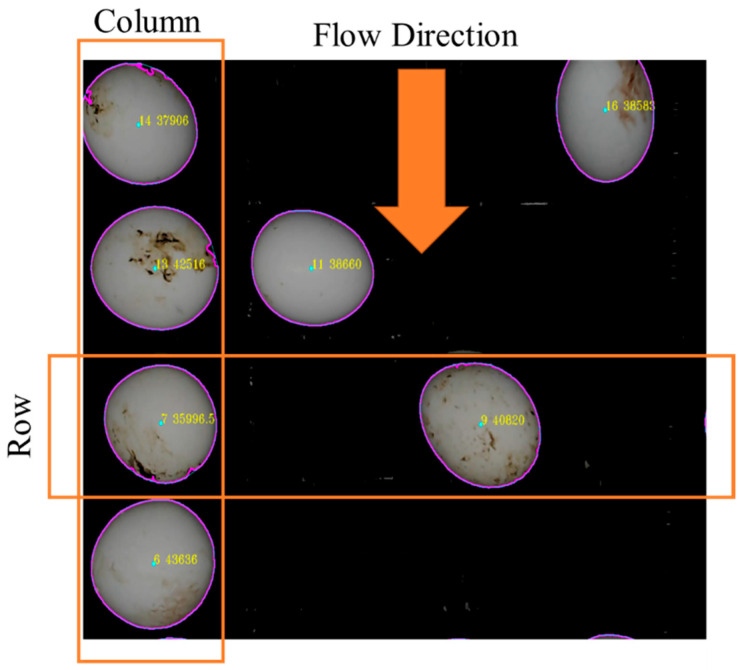
Eggs on the industrial movement halls.

**Figure 2 sensors-23-00117-f002:**
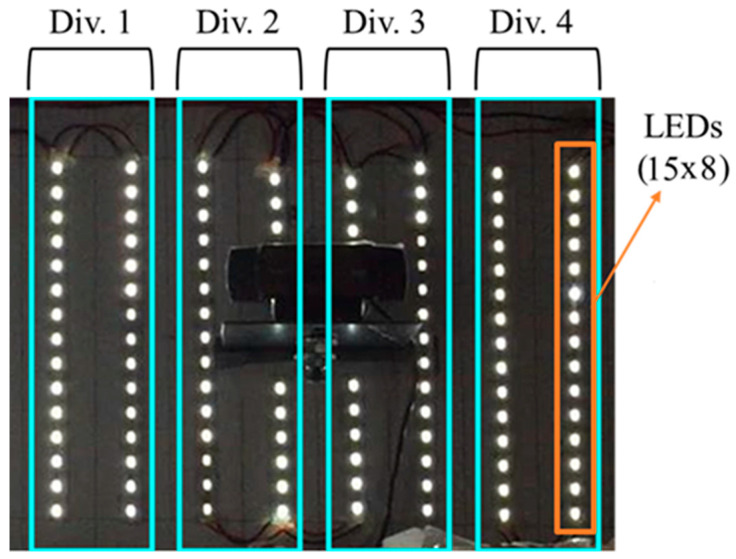
Light sources and imaging device.

**Figure 3 sensors-23-00117-f003:**
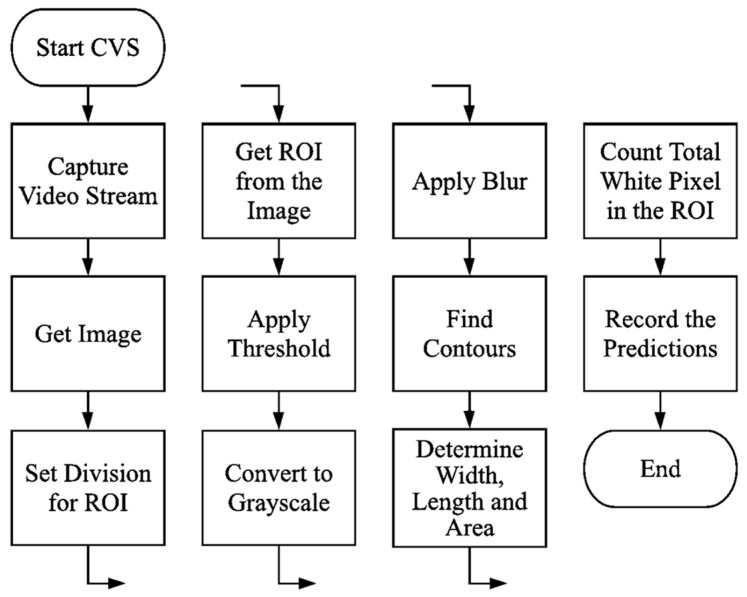
Flowchart of the image processing procedure.

**Figure 4 sensors-23-00117-f004:**
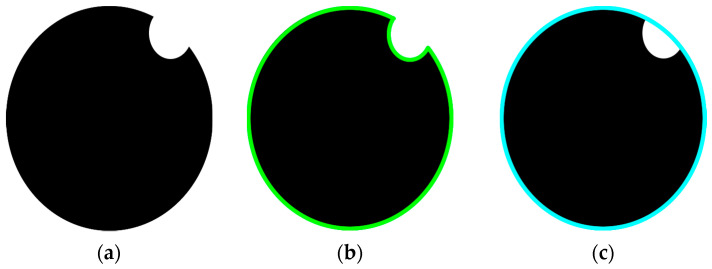
Egg circulating procedure, (**a**) actual, (**b**) contoured, (**c**) minimum enclosing circle.

**Figure 5 sensors-23-00117-f005:**
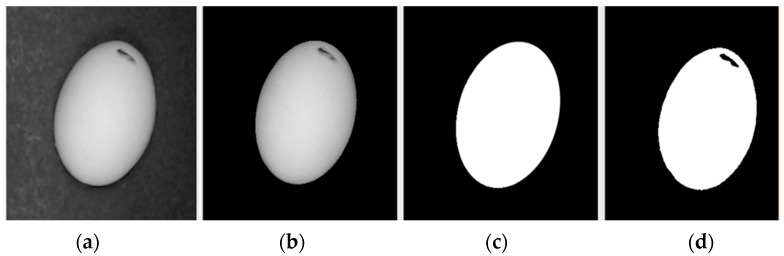
Egg detecting procedure; (**a**) actual image, (**b**) threshold of the grayscale image, (**c**) mask, (**d**) detected egg.

**Figure 6 sensors-23-00117-f006:**
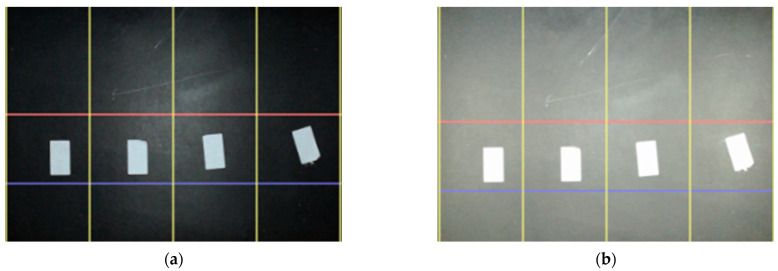
Illuminating the measurement area using independent (**a**) single and (**b**) multiple light sources.

**Figure 7 sensors-23-00117-f007:**
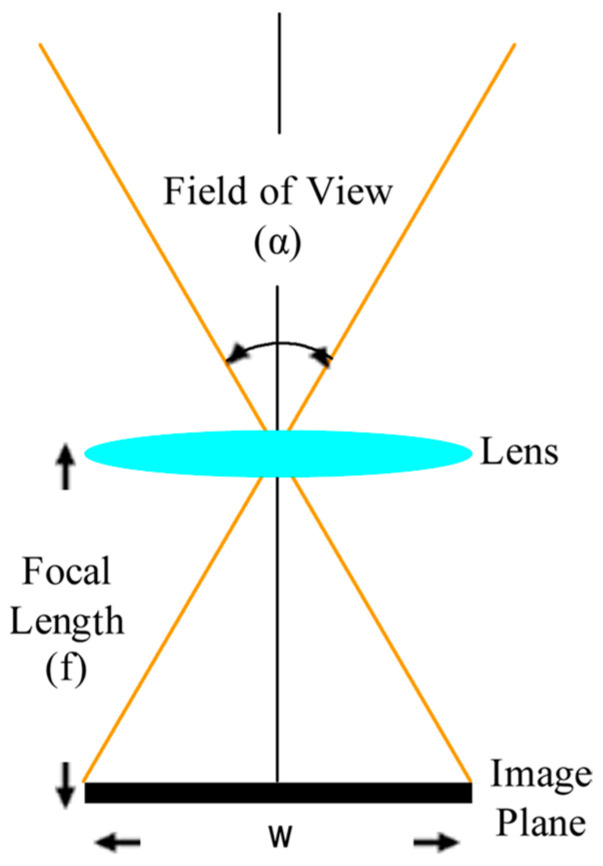
Relationship between focal length, *FoV* and sensor size.

**Figure 8 sensors-23-00117-f008:**
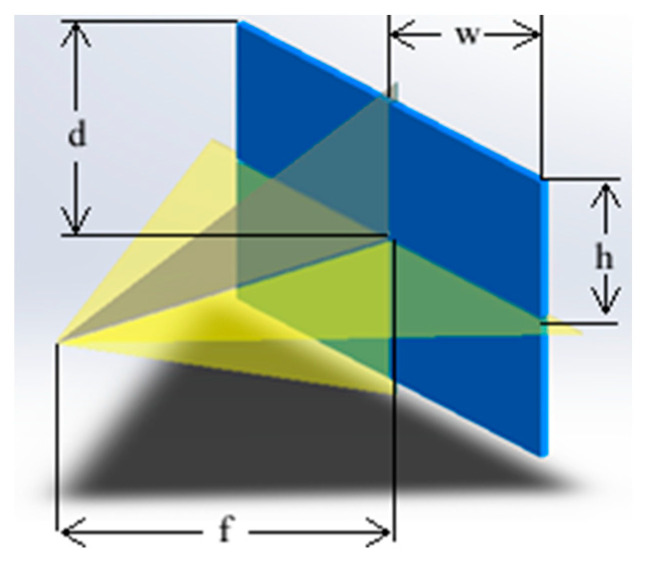
Isometric view of *FoV* (yellow) and reference rectangular frame (blue).

**Figure 9 sensors-23-00117-f009:**
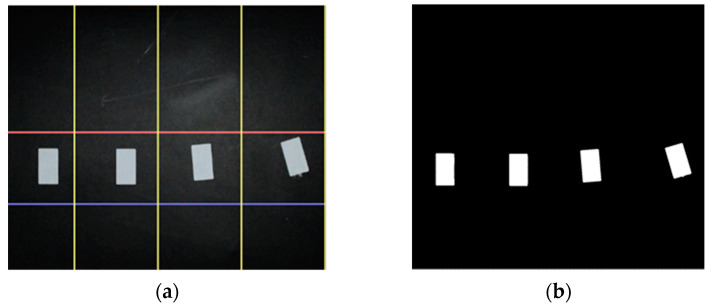
The original image (**a**) and processed image (**b**) of the four identical targets in flow lines.

**Figure 10 sensors-23-00117-f010:**
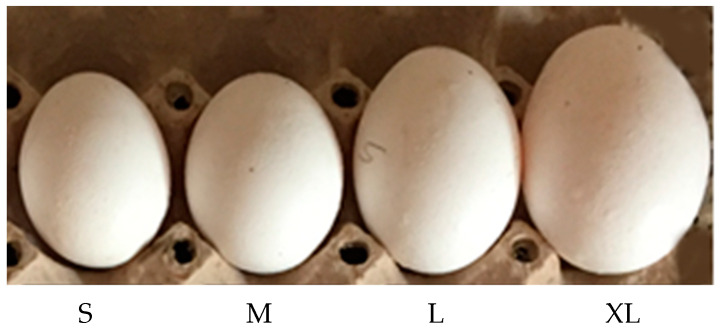
Sample eggs graded concerning to their weights.

**Figure 11 sensors-23-00117-f011:**
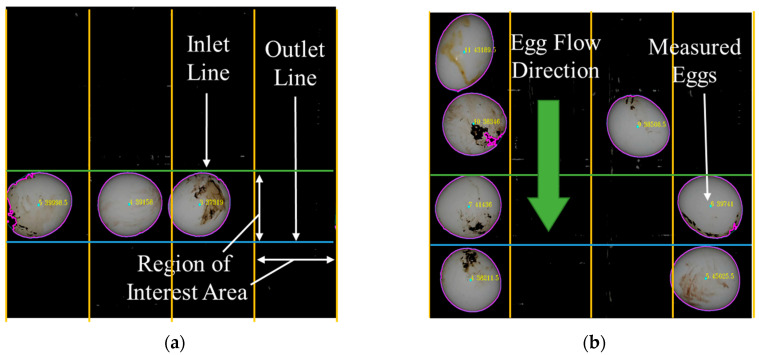
An image of detected eggs on the movement halls (**a**,**b**).

**Figure 12 sensors-23-00117-f012:**
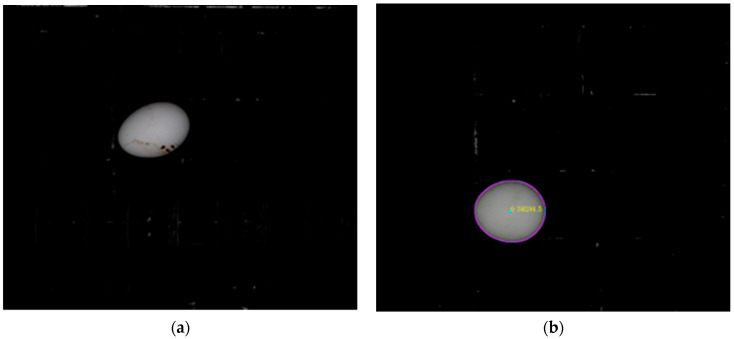
Single-line experiment (**a**), detected egg (**b**), multi-line experiment (**c**), and detected eggs (**d**).

**Figure 13 sensors-23-00117-f013:**
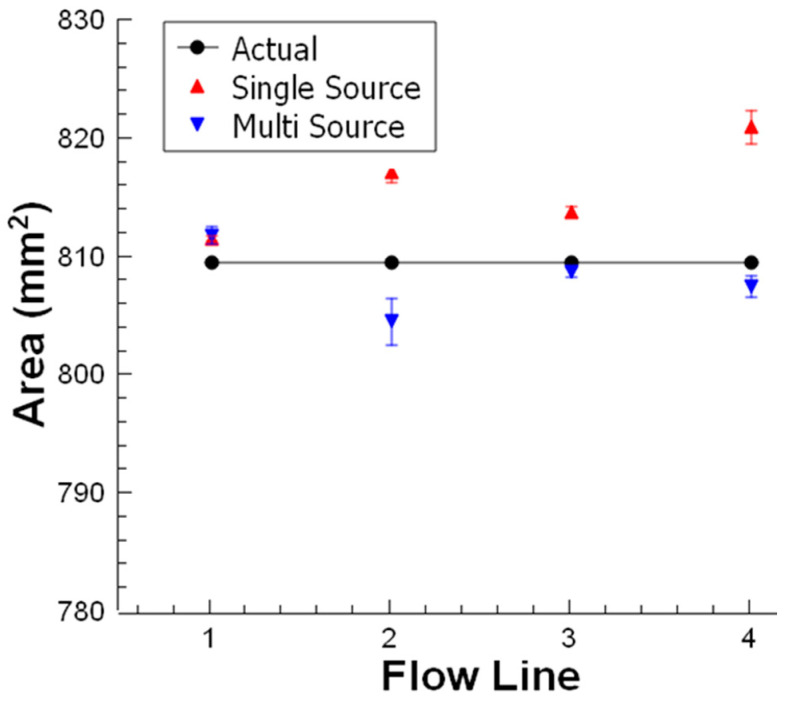
The effect of single and multi-light source.

**Figure 14 sensors-23-00117-f014:**
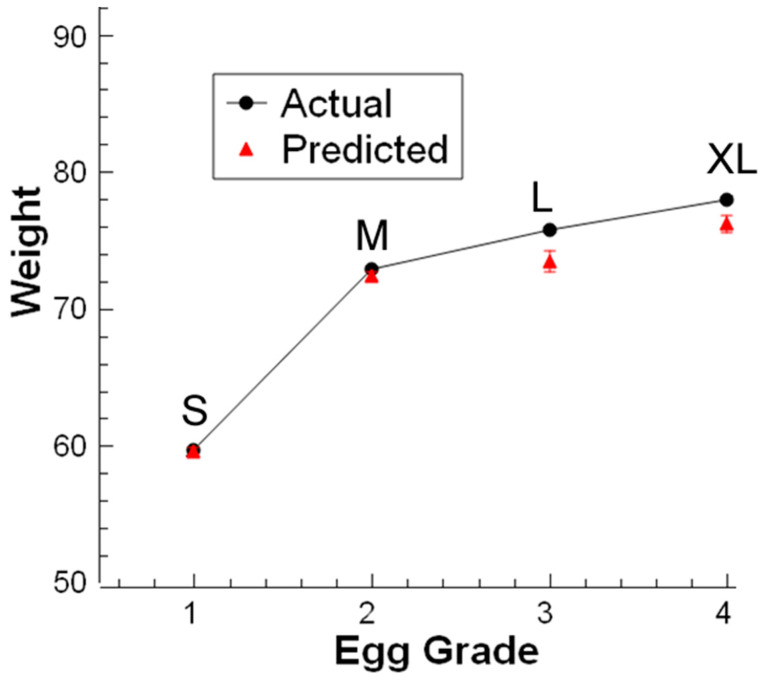
Four graded eggs weight measurements on a single-line flow.

**Figure 15 sensors-23-00117-f015:**
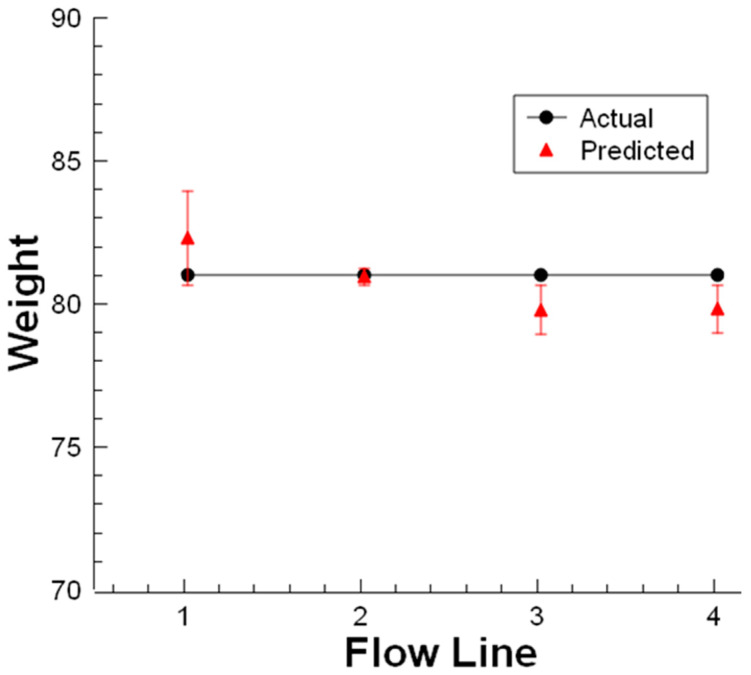
One egg weight prediction on multi-line flow.

**Figure 16 sensors-23-00117-f016:**
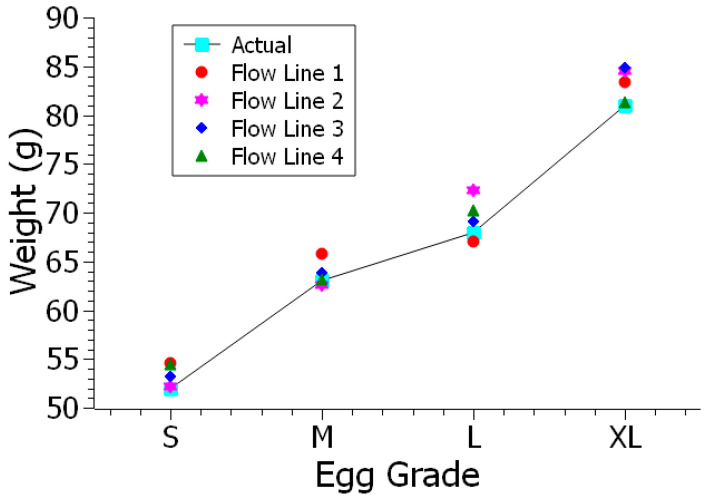
Multi-line flow experiment with multiple eggs.

**Table 1 sensors-23-00117-t001:** European egg size standard [15,16].

Grade	Weight	Classification
1	>73	XL—Extra
2	>63 and ≤73	L—Large
3	>53 and ≤63	M—Medium
4	<53	Small

**Table 2 sensors-23-00117-t002:** Elapsed time of the image processing cycle.

Process	Time (sec)
Frame capture	0.0025324
+Select ROI +GrayScale +Blur +Gaussian Filter	0.0035324
+Find Contors	0.0045324
+Predict Egg Weights	0.0055324
**One Cycle**	**0.092**

## Data Availability

Not applicable.

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
