# Peer review of "A Multi-Flow Production Line for Sorting of Eggs Using Image Processing"

_sensors, 2022, doi:10.3390/s23010117_

Round 1

Reviewer 1 Report

 Review Comments

The authors explained the image processing technology that was used to classify chicken eggs on an industrial roller conveyor line in real time. A color camera was used to acquire images in an illumination cabinet on a motorized roller conveyor while eggs are moving on the movement halls. The system successfully operated for the grading of eggs in the industrial multi-flow production line in real time. However, the following corrections can be considered by the authors to further improve the quality of the manuscript.

 I have some corrections and suggestions below:-

1. Novelty of the work can be highlighted better. What novelty is established in this work compared to existing works? The novelty of the work can be added at the end of Introduction section.

2. Literature survey missing. Various Papers based on the current state of artwork must be included in a literature survey

3. Why were LEDs (4 W) total of 210 pieces utilized in the imaging device? Justify?

4. Which algorithms have been utilized for object detection and classification?

5. What are the conditions for illumination and properties of illumination in real-time industrial cases utilized?

6. Results with respect to real-time analysis like FPS must be tested and verified.

7. The computational complexity of the algorithm must be discussed. Also, compare the proposed method in terms of computational complexity.

8. Limitations of the proposed work and future work can be added and discussed.

9. Results based on broken, expert evaluation and machine sorting criteria must be added and tabulated the results in terms of accuracy and other parameters.

10. How many eggs have been considered at a time and also include based on the consideration of egg.

11. How image processing algorithms have been applied. Authors must explain the algorithm with some more results and also validate the results to claim the novel part of the research.

12. Results shown in the graph can be tabulated in terms of various performance parameters.

13. Results with respect to some more images must be tested and verified. Results based on timing analysis must be added and compared.

14. Robustness of the proposed algorithm must be discussed and analyzed graphically.

Reviewer 2 Report

I have the following concerns.

1. There is no clear statement of the problem. What is the scientific novelty of your method. The use of the OpenCV library to select contours is not a novelty.

2. The image of the egg depends on the angle at which it is illuminated. How to eliminate this dependence.

3. It is not clear what methods of processing, detection and classification were used. They should be described in the text of the article.

4. It is necessary to give comparative estimates of accuracy and recognition time in comparison with other methods in the table. After all, the accuracy and time of deciding to which class the recognized egg belongs is crucial.

5. References should include articles for 2021-2022 that would confirm the relevance of the problem.

6. This information is missing in the conclusions about machine learning methods and in the text of the article.

Round 2

Reviewer 1 Report

All my concerns and comments has been added and modified satisfactory level. I accept it in current from.